# Estimating Transition Matrix with Diffusion Models for Instance-Dependent Label Noise

## Abstract

Learning with noisy labels is a common problem in weakly supervised learning, where the transition matrix approach is a prevalent method for dealing with label noise. It estimates the transition probabilities from a clean label distribution to a noisy label distribution and has garnered continuous attention. However, existing transition matrix methods predominantly focus on class-dependent noise, making it challenging to incorporate feature information for learning instance-dependent label noise. This paper proposes the idea of using diffusion models for estimating transition matrix in the context of instance-dependent label noise. Specifically, we first estimate grouped transition matrices through clustering. Then, we introduce a process of adding noise and denoising with the transition matrix, incorporating features extracted by unsupervised pre-trained models. The proposed method enables the estimation of instance-dependent transition matrix and extends the application of transition matrix method to a broader range of noisy label data. Experimental results demonstrate the significant effectiveness of our approach on both synthetic and real-world datasets with instance-dependent noise. The code will be open sourced upon acceptance of the paper.

## 1 Introduction

For classification problems with given labels, deep neural networks have demonstrated significant improvements compared to traditional methods in recent years [25]. The efficacy of deep neural networks heavily relies on the accuracy of the labels. Directly incorporating polluted erroneous labels into network learning can result in the network fitting the noise, potentially severely impacting the predictive performance of the network [8]. However, in reality, obtaining accurate annotated data can be prohibitively expensive, and a substantial amount of data comes from the Internet or is annotated by non-expert annotators, inevitably containing noisy labels. Therefore, researching and promoting methods to mitigate the damage to models and make them more robust in the face of label noise data is a highly worthwhile problem to investigate, known as the problem of learning with noisy labels [23, 10, 34, 1].

Different approaches have been proposed to address the problem of label noise. One category [31, 22] involves the design of specialized loss functions or network structures to enhance the model's robustness against noisy labels. Another major category focuses on sample selection [2, 10, 14], where samples are partitioned into a set of clean samples and a set of contaminated noisy samples based on the magnitude of the loss or the similarity of extracted features. The labels of the noisy samples are then modified or their weights are reduced, followed by learning using semi-supervised methods. Sample selection methods are currently mainstream and have achieved promising results. However, the selection process relies heavily on intuition and lacks theoretical support. Additionally, the sample selection procedure is often complex and computationally intensive. In contrast, another

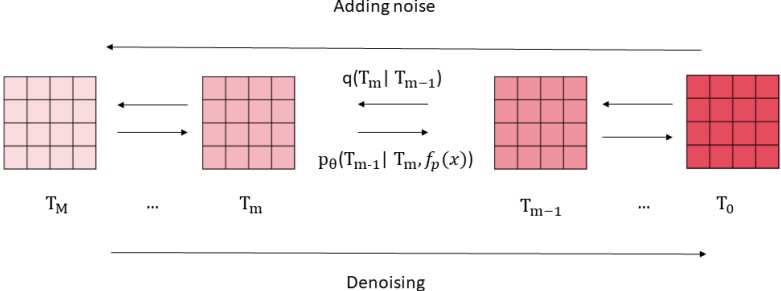

Figure 1: Diffusion Model for Transition Matrix.

significant category of methods is the transition matrix method [34, 17, 12, 42], which estimates the transition probabilities from the clean label distribution to the noisy label distribution. This class of methods reveals the generation process of noisy labels and exhibits statistical consistency, often accompanied by theoretical analyses as methodological support. As a result, they have garnered continuous attention and occupy an important position in various algorithms for learning with noisy labels.

In transition matrix methods, accurate estimation of the transition matrix is crucial. If an accurate estimation of the transition matrix can be obtained, along with the observed data for estimating the posterior distribution of the noisy labels, it is possible to infer the distribution of clean labels for neural network learning. Previous transition matrix methods [34, 17, 39] have mainly focused on class-dependent label noise, where a single transition matrix is estimated for all samples, which is typically straightforward. However, for instance-dependent label noise and complex real-world data, the label transition probabilities for each sample are not entirely identical. The transition matrix often depends on the specific features of individual samples, requiring the estimation of a separate transition matrix for each sample. However, in most cases, a single observed label corresponds to each sample in the dataset, making it an identifiability problem to estimate a separate transition matrix for each sample [20]. Although some methods [33, 41, 15] have utilized separate small networks to generate the transition matrix or divided the data into groups to transform it into a grouped class-dependent scenario, there still exist significant estimation errors and a lack of incorporating features effectively into the estimation of the transition matrix.

To better incorporate the feature information of images into the estimation of the transition matrix, this work employs conditional diffusion models. The diffusion model originates from generative models and has been widely applied in various computer vision tasks in recent years [36, 7], showing remarkable results. The proposed method revolves around the core idea of replacing image samples in the original diffusion process with a transition matrix. The matrix undergoes a process of adding noise and denoising, where the denoising step incorporates the sample features extracted by a pre-trained model as conditions. This generates a feature-dependent transition matrix. The constructed diffusion module is illustrated in Figure 1. Additionally, considering the assumption that instance-dependent label noise is usually correlated with features [6], clustering methods are utilized at the feature level to group samples. Preliminary estimations of the transition matrices are obtained for each group, which are then incorporated into the diffusion module for learning. The overall framework of the method is depicted in Figure 2.

The subsequent sections are organized as follows. Section 2 presents an in-depth review of the relevant works. In Section 3, we introduce our proposed model framework. Section 4 outlines the experimental analysis conducted on diverse synthetic and real-world noisy datasets, along with comparisons against other existing methods. Finally, we provide concluding in Section 5. The primary contributions of this paper can be summarized as follows:

- We propose a method that utilizes diffusion models to add noise and denoise on the transition matrix, incorporating image features extracted through pre-trained encoder.

- By combining the transition matrix-based diffusion model with feature-based clustering, we establish a framework capable of addressing instance-dependent label noise problems.

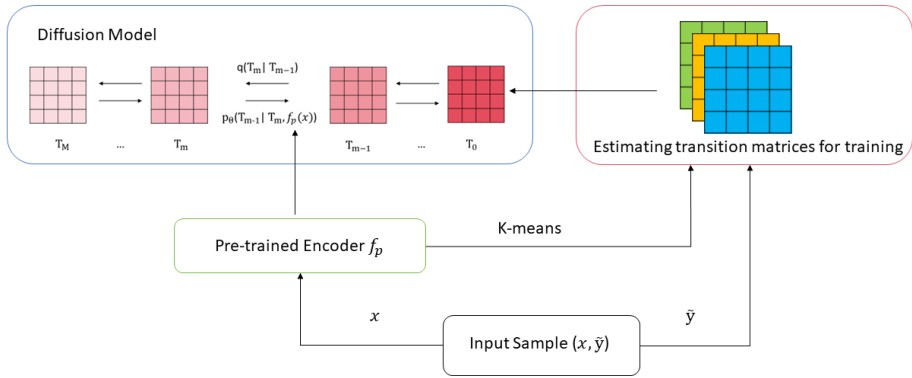

Figure 2: The overall framework of DTM.

- Our method demonstrates significant improvements over other transition matrix methods on both synthetic and real-world noisy datasets, and it achieves comparable performance to state-of-the-art methods.

## 2   Related Works

### 2.1   Transition Matrix Methods

Most previous methods for estimating transition matrix in the presence of label noise have primarily focused on class-dependent noise scenarios, simplifying the estimation process. Methods such as [24, 34] assume the existence of anchor points to identify the transition matrix. [17] and [39] introduce different regularization techniques to relax the anchor point assumption. Additionally, [26, 38] apply techniques such as meta-learning to estimate the transition matrix, but these approaches may require more clean data and computational resources. While these methods are effective for handling class-dependent label noise, they are not suitable for instance-dependent noise or real-world noisy data.

However, estimating an individual transition matrix for each sample without additional assumptions or multiple noisy labels is infeasible [20]. To approximate the estimation of the instance-dependent transition matrix, [9] utilize an adaptation layer that estimates the transition matrix based on the output of each sample. [37] employs a separate network to estimate the transition matrix based on Bayesian labels. Some methods, such as [33, 30, 41], employ clustering to learn part-dependent or group-dependent matrices, which can be viewed as a compromise between instance-dependent and class-dependent methods. Other approaches, including [6, 12], utilize the similarity in the feature space to aid in learning the transition matrix. Although these instance-dependent transition matrix methods achieve identifiability through specialized treatments, they have not effectively utilized feature information in the learning process, resulting in errors in estimating feature-dependent transition matrices.

### 2.2   Diffusion Models

Diffusion models, as generative models, have played a significant role in computer vision [36, 7]. Prominent examples include DDPM [11], DDIM [27], score matching methods [28], and methods based on stochastic differential equations [29]. Diffusion models and their variants have been applied to various computer vision tasks such as image generation, image-to-image translation, text-to-image generation, among others. However, their application to the problem of label noise is relatively novel. To the best of our knowledge, only one existing work [3] has utilized diffusion models for addressing this problem. However, this work treats labels as the output of the diffusion model, which limits their expressive power due to the low dimension of the labels. Moreover, it overly relies on directly incorporating image features as conditions in the label generation process, which depends heavily on

pre-trained models and may not be as reasonable as incorporating them into the transition matrix that reveals the process of noise generation. Experimental results also support this perspective.

## 3 Method

In this section, we present the definitions of symbols and introduce our method of using **D**iffusion models to construct the **T**ransition **M**atrix (DTM).

### 3.1 Preliminaries

Let $\mathcal{X} \subset \mathbb{R}^d$ be the input image space, $\mathcal{Y} = \{1, 2, \cdots, C\}$ be the label space, where $C$ is the number of classes. Random variables $(X, Y), (X, \tilde{Y}) \in \mathcal{X} \times \mathcal{Y}$ denote the underlying data distributions with true and noisy labels respectively. In general, we can not observe the latent true data samples $\mathbb{D} = \{(\boldsymbol{x}_i, y_i)\}_{i=1}^{N}$, but can only obtain the corrupted data $\tilde{\mathbb{D}} = \{(\boldsymbol{x}_i, \tilde{y}_i)\}_{i=1}^{N}$, where $\tilde{y} \in \mathcal{Y}$ is the noisy label corrupted from the true label $y$, while denote corresponding one-hot label as $\boldsymbol{y}$ and $\tilde{\boldsymbol{y}}$.

Transition matrix methods use a matrix $\boldsymbol{T}(\boldsymbol{x}) \in [0, 1]^{C \times C}$ to represent the probability from clean label to noisy label, where the $ij$-th entry of the transition matrix is the probability that the instance $\boldsymbol{x}$ with the clean label $i$ corrupted to a noisy label $j$. The matrix satisfies the requirement that the sum of each row $\sum_{j=1}^{C} \boldsymbol{T}_{ij}(\boldsymbol{x})$ is 1, and usually has the requirement for $\boldsymbol{T}_{ii}(\boldsymbol{x}) > \boldsymbol{T}_{ij}(\boldsymbol{x}), \forall j \neq i$. Let $P(\boldsymbol{Y}|X = \boldsymbol{x}) = [P(Y = 1|X = \boldsymbol{x}), \cdots, P(Y = C|X = \boldsymbol{x})]^{\top}$ be the clean class-posterior probability and $P(\tilde{\boldsymbol{Y}}|X = \boldsymbol{x}) = [P(\tilde{Y} = 1|X = \boldsymbol{x}), \cdots, P(\tilde{Y} = C|X = \boldsymbol{x})]^{\top}$ be the noisy class-posterior probability, the formula can be write as:

$$P(\tilde{\boldsymbol{Y}}|X = \boldsymbol{x}) = \boldsymbol{T}(\boldsymbol{x})^{\top} P(\boldsymbol{Y}|X = \boldsymbol{x}). \tag{1}$$

By estimating the transition matrix and the noisy class-posterior probability, the clean class-posterior probability can be inferred by

$$P(\boldsymbol{Y}|X = \boldsymbol{x}) = \boldsymbol{T}(\boldsymbol{x})^{-\top} P(\tilde{\boldsymbol{Y}}|X = \boldsymbol{x}), \tag{2}$$

where the symbol $-\top$ denotes the transpose of the inverse matrix.

The majority of existing methods [24, 10, 17] focus on studying the class-dependent and instance-independent transition matrix, i.e., $\boldsymbol{T}(\boldsymbol{x}) \equiv \boldsymbol{T}$ for $\forall \boldsymbol{x}$. However, these methods are not applicable to instance-dependent noise scenarios where the transition matrix $\boldsymbol{T}(\boldsymbol{x})$ varies with respect to the input $X$. The main focus of our work is to utilize the feature information from input images to construct a instance-dependent transition matrix $\boldsymbol{T}(\boldsymbol{x})$.

### 3.2 Diffusion Model for Transition Matrix

We adopt the classic DDPM model [11] from diffusion models as a reference to perform noise addition and denoising on the transition matrix. The diagram is illustrated in Figure 1.

For the forward diffusion process beginning with transition matrix $\boldsymbol{T}_0 \sim q(\boldsymbol{T})$, the process of gradually adding noise is obtained according to the following Markov process:

$$q\left(\boldsymbol{T}_m \mid \boldsymbol{T}_{m-1}\right) = \mathcal{N}\left(\boldsymbol{T}_m; \sqrt{1 - \beta_m}\boldsymbol{T}_{m-1}, \beta_m \mathbf{I}\right), \tag{3}$$

for $m = 1, 2, \cdots, M$, where we use $M$ to replace $T$, which is usually used in other diffusion models, in above equation for distinguishing from the symbol of transition matrix $\boldsymbol{T}$.

We aim to make the distribution of $q(\boldsymbol{T}_M)$ approach a standard normal distribution $\mathcal{N}(0, \mathbf{I})$ and through $\boldsymbol{T}_M$ to conduct the reverse denoising process by fitting a neural network $\boldsymbol{\mu}_\theta$ to fit the disttibution:

$$p_\theta\left(\boldsymbol{T}_{m-1} \mid \boldsymbol{T}_m\right) = \mathcal{N}\left(\boldsymbol{T}_{m-1}; \boldsymbol{\mu}_\theta\left(\boldsymbol{T}_m, \boldsymbol{x}, f_p, m\right), \tilde{\beta}_m \mathbf{I}\right), \tag{4}$$

where define $\tilde{\beta}_m = \frac{1 - \bar{\alpha}_{m-1}}{1 - \bar{\alpha}_m}\beta_m, \alpha_m = 1 - \beta_m, \bar{\alpha}_m = \prod_{i=1}^{m} \alpha_i$. The $f_p$ in equation (4) denotes the pre-trained encoder for feature extraction.

The diffusion model can be learned by optimizing the evidence lower bound:

$$\mathcal{L}_{\text{ELBO}} = \mathbb{E}_q \left[ \mathcal{L}_M + \sum_{m>1}^{M} \mathcal{L}_{m-1} + \mathcal{L}_0 \right],$$ (5)

where

$$\begin{aligned}
\mathcal{L}_0 &= -\log p_\theta \left( \boldsymbol{T}_0 \mid \boldsymbol{T}_1 \right), \\
\mathcal{L}_{m-1} &= D_{\text{KL}} \left( q \left( \boldsymbol{T}_{m-1} \mid \boldsymbol{T}_m, \boldsymbol{T}_0 \right) \| p_\theta \left( \boldsymbol{T}_{m-1} \mid \boldsymbol{T}_m \right) \right), \\
\mathcal{L}_M &= D_{\text{KL}} \left( q \left( \boldsymbol{T}_M \mid \boldsymbol{T}_0 \right) \| p_\theta \left( \boldsymbol{T}_M \right) \right).
\end{aligned}$$ (6)

Similar to the derivation and simplification process of DDPM, when a pre-trained encoder $f_p$ is provided along with the training data incorporating the initial transition matrix $\boldsymbol{T}$, the learning algorithm for the diffusion model is presented in Algorithm 1.

---

**Algorithm 1** Diffusion Model for Transition Matrix

---

**Input:** Training data $\{\boldsymbol{x}_i, \boldsymbol{T}_i\}_{i=1}^{N}$, pre-trained encoder $f_p$.
**while** not converged **do**
  Sample $(\boldsymbol{x}_0, \boldsymbol{T}_0)$ from data
  Sample $m \sim \{1, \cdots, M\}$
  Sample noise $\boldsymbol{\epsilon} \sim \mathcal{N}(0, \mathbf{I})$
  Take gradient descent step on the loss:

$$\nabla_\theta \left\| \boldsymbol{\epsilon} - \boldsymbol{\epsilon}_\theta \left( \sqrt{\bar{\alpha}_m} \boldsymbol{T}_0 + \sqrt{1 - \bar{\alpha}_m} \boldsymbol{\epsilon}, \boldsymbol{x}_0, f_p, m \right) \right\|^2$$

**end while**

---

Next, for each image $\boldsymbol{x}$, we can sample the corresponding transition matrix $\boldsymbol{T}(\boldsymbol{x})$ as shown in Algorithm 2.

---

**Algorithm 2** Sample for Transition Matrix

---

Sample $\boldsymbol{T}_M \sim \mathcal{N}(0, \mathbf{I})$
**for** $m = M, \cdots, 1$ **do**
  $\boldsymbol{z} \sim \mathcal{N}(\mathbf{0}, \mathbf{I})$ if $t > 1$, else $\boldsymbol{z} = \mathbf{0}$
  $\boldsymbol{T}_{m-1} = \frac{1}{\sqrt{\alpha_m}} \left( \boldsymbol{T}_m - \frac{1 - \alpha_m}{\sqrt{1 - \bar{\alpha}_m}} \boldsymbol{\epsilon}_\theta \left( \boldsymbol{T}_m, \boldsymbol{x}, f_p, m \right) \right) + \sigma_m \boldsymbol{z}$
**end for**
**Output:** $\boldsymbol{T}_0$

---

### 3.3 Feature-Dependent Framework

From Algorithm 1, it can be observed that there are two components of the diffusion process that need to be provided in advance: the pre-trained encoder $f_p$ and the initial input $\boldsymbol{T}(\boldsymbol{x})$.

The pre-trained encoder $f_p$ can be obtained through self-supervised learning or directly using the large model like CLIP. In our experiments, we employ the commonly used SimCLR [4] method in contrastive learning as the feature extraction model.

On the other hand, the part involving the transition matrix $\boldsymbol{T}(\boldsymbol{x})$ used for learning the diffusion model is also related to the pre-trained encoder $f_p$. Based on the assumption that the noise transition probability depends on image features, we adopt a group-dependent transition matrix as the initial input. We perform clustering algorithms at the feature extraction level $f_p(\boldsymbol{x})$, using the K-means method in our experiments, to group the image data. Then, based on the method VolMinNet [17], we train class-dependent transition matrices for each group and obtain the initial transition matrix $\boldsymbol{T}(\boldsymbol{x})$ for each image $\boldsymbol{x}$, which is then used as input in Algorithm 1. It is worth to note that the initial $\boldsymbol{T}(\boldsymbol{x})$ used as input for the diffusion process does not require different for each $\boldsymbol{x}$. However, the denoising process of the diffusion model will further incorporate the feature information into the learning of the transition matrix.

After obtaining the instance-dependent estimated transition matrix $\boldsymbol{T}(\boldsymbol{x})$, the neural network can be learned to fit the clean label distribution by the loss function:

$$\mathcal{L} = \frac{1}{N} \sum_{i=1}^{N} \ell \left( \boldsymbol{T}(\boldsymbol{x}_i)^\top f_\phi(\boldsymbol{x}_i), \tilde{\boldsymbol{y}}_i \right), \tag{7}$$

where $f_\phi(\cdot) : \mathcal{X} \to \Delta^{C-1}$ ($\Delta^{C-1} \subset [0,1]^C$ is the $C$-dimensional simplex) is a differentiable function represented by a neural network with parameters $\phi$ and $\ell$ is a loss function usually using cross-entropy (CE) loss.

The schematic diagram of the proposed framework is shown in Figure 2, and the pseudocode is presented in Algorithm 3.

---

**Algorithm 3** A framework of DTM

---

**Input:** Training set $\{(\boldsymbol{x}_i, \boldsymbol{y}_i)\}_{i=1}^N$, pre-trained encoder $f_p$, diffusion model $\epsilon_\theta$, classification neural network $f_\phi$.
1: Utilize input data to train $f_p$ or directly utilizing $f_p$ to extract features.
2: Perform K-means on feature space and estimate the transition matrix for each group to get data $\{\boldsymbol{x}_i, \boldsymbol{T}_i\}_{i=1}^N$.
3: Train the diffusion model $\epsilon_\theta$ with Algorithm 1.
4: Sample instance-dependent train matrix $\boldsymbol{T}(\boldsymbol{x})$ for any input image $\boldsymbol{x}_i$ with Algorithm 2.
5: Update the parameters of the classification network by incorporating the transition matrix $\boldsymbol{T}(\boldsymbol{x}_i)$ into equation (7).

**Output:** Network parameters $\phi$.

---

### 3.4 Matrix Transformation

Considering that the transition matrix typically require the sum of each row $\sum_{j=1}^C \boldsymbol{T}_{ij}(\boldsymbol{x})$ is 1, and for $\boldsymbol{T}_{ii}(\boldsymbol{x}) > \boldsymbol{T}_{ij}(\boldsymbol{x}), \forall j \neq i$, we employ a transformation during the update learning process in our practical experiments.

We utilize a $C \times C$ weight matrix $\boldsymbol{W} = (w_{ij})$ to assist in the process. Denote matrix $\boldsymbol{A}$ as $\boldsymbol{A}_{ii} = 1 + \sigma(w_{ii})$ for all $i \in \{1, 2, \ldots, C\}$ and $\boldsymbol{A}_{ij} = \sigma(w_{ij})$ for all $i \neq j$ where $\sigma$ is the sigmoid function. Then we do the normalization $\boldsymbol{T}_{ij} = \frac{\boldsymbol{A}_{ij}}{\sum_{k=1}^C \boldsymbol{A}_{kj}}$ to get the transition matrix $\boldsymbol{T}$.

Through this transformation, we ensure that the learned transition matrix has row sums equal to 1 and that the diagonal elements are the largest in each row. In practical experiments, we apply the diffusion modeling discussed in subsection 3.2 to the matrix $\boldsymbol{W}$, and then transform it into the transition matrix $\boldsymbol{T}$ for application. To simplify the notation, we uniformly use the term of transition matrix $\boldsymbol{W}$ to represent it, unless it leads to singularity.

## 4 Experiments

In this section, we present experimental findings to showcase the effectiveness of our proposed method compared to other methods. We evaluate our approach on both synthetic instance-dependent noisy datasets and real-world noisy datasets.

### 4.1 Datasets

We conduct experiments on following image classification datasets: CIFAR-10 and CIFAR-100 [13], CIFAR-10N and CIFAR-100N [32], Clothing1M [35], Webvision and ILSVRC12 [16]. Among them, CIFAR-10 and CIFAR-100 both have $32 \times 32 \times 3$ color images including 50,000 training images and 10,000 test images. CIFAR-10 has 10 classes while CIFAR-100 has 100 classes. We generate instance-dependent noisy data on CIFAR-10 and CIFAR-100 with noise rates ranging from 10% to 50%, following the same generation method as in [33]. CIFAR-10N has three annotated labels, namely Random1, Random 2 and Random 3. The "Aggregate" is the aggregation of three noisy labels by majority voting, and the "Worst" is the dataset with the worst case. For CIFAR-100N, each

image contains a coarse label and a fine label given by a human annotator. Clothing1M is a real-world dataset consisting of 1 million training images, consisting of 14 categories. WebVision contains 2.4 million images crawled from the websites using the 1,000 concepts in ImageNet ILSVRC12, but only the first 50 classes of the Google image subset are used in our experiments. For the validation set selection in our BTR method, we randomly sampled 10 samples from each observed class for each dataset to form the validation set, while the remaining samples were used for the training set.

## 4.2 Experimental Setup

For the pre-trained model, we employ the commonly used SimCLR model [4] from contrastive learning, which directly performs self-supervised learning on input images without utilizing additional datasets. For the diffusion model, we follow the setup similar to DDPM [11] to set $\beta_1 = 10^{-4}, \beta_M = 0.02$ and utilize a similar U-Net network architecture but we reduce the $M$ from 1000 to 10 to accelerate the learning process. As for the classification network, it may vary depending on the specific dataset. More specifically, for CIFAR-10/10N, we use ResNet-18 as the backbone network with batch size 128 and learning rate 0.05. For CIFAR-100/100N, we use ResNet-34 network with batch size 128, learning rate 0.02. For clothing1M, we use a ResNet-50 pre-trained with 10 epochs, batch size 64, learning rate 0.002 for network and divided by 10 after the 5th epoch. We use InceptionResNetV2 network on Webvision, with 100 epochs, batch size 32, learning rate 0.02 for network and divided by 10 after the 30th and 60th epoch. For clustering, we utilize the K-means method, where the number of clusters is set to 10 times the number of classes in the datasets. For the initialization of transition matrix, the update method and setting are consistent with [17]. While the updates for other parameters are performed using the stochastic gradient descent optimization method.

Table 1: Test accuracy with instance-dependent noise on CIFAR-10/100.

| | CIFAR-10 | | | | |
| --- | --- | --- | --- | --- | --- |
| | IDN-10% | IDN-20% | IDN-30% | IDN-40% | IDN-50% |
| CE | 88.86±0.23 | 86.93±0.17 | 82.42±0.44 | 76.68±0.23 | 58.93±1.54 |
| VolMinNet | 89.97±0.57 | 87.01±0.64 | 83.80±0.67 | 79.52±0.83 | 61.90±1.06 |
| PeerLoss | 90.89±0.07 | 89.21±0.63 | 85.70±0.56 | 78.51±1.23 | 59.08±1.05 |
| BLTM | 90.45±0.72 | 88.14±0.66 | 84.55±0.48 | 79.71±0.95 | 63.33±2.75 |
| PartT | 90.32±0.15 | 89.33±0.70 | 85.33±1.86 | 80.59±0.41 | 64.58±2.86 |
| MEIDTM | 92.91±0.07 | 92.26±0.25 | 90.73±0.34 | 85.94±0.92 | 73.77±0.82 |
| SOP | 93.58±0.31 | 93.07±0.45 | 92.42±0.43 | 89.83±0.77 | 82.52±0.97 |
| CC | 95.24±0.20 | 93.68±0.12 | 93.31±0.46 | **94.97±0.09** | 91.19±0.34 |
| LRA | 95.87±0.42 | 94.70±0.28 | 93.79±0.40 | 92.72±0.29 | 90.95±0.43 |
| DTM | **96.45±0.17** | **95.90±0.21** | **95.14±0.20** | 94.82±0.31 | **92.04±0.42** |
| | CIFAR-100 | | | | |
| | IDN-10% | IDN-20% | IDN-30% | IDN-40% | IDN-50% |
| CE | 66.55±0.23 | 63.94±0.51 | 61.97±1.16 | 58.70±0.56 | 56.63±0.69 |
| VolMinNet | 67.78±0.62 | 66.13±0.47 | 61.08±0.90 | 57.35±0.83 | 52.60±1.31 |
| PeerLoss | 65.64±1.07 | 63.83±0.48 | 61.64±0.67 | 58.30±0.80 | 55.41±0.28 |
| BLTM | 68.42±0.42 | 66.62±0.85 | 64.72±0.64 | 59.38±0.65 | 55.68±1.43 |
| PartT | 67.33±0.33 | 65.33±0.59 | 64.56±1.55 | 59.73±0.76 | 56.80±1.32 |
| MEIDTM | 69.88±0.45 | 69.16±0.16 | 66.76±0.30 | 63.46±0.48 | 59.18±0.16 |
| SOP | 74.09±0.52 | 73.13±0.46 | 72.14±0.46 | 68.98±0.58 | 64.24±0.86 |
| CC | 80.52±0.22 | 79.61±0.19 | 77.34±0.31 | 76.58±0.25 | 72.68±0.36 |
| LRA | 81.20±0.16 | 80.53±0.29 | 78.22±0.19 | 76.55±0.31 | 72.97±0.51 |
| DTM | **82.96±0.25** | **82.04±0.32** | **80.87±0.45** | **78.56±0.60** | **74.85±0.56** |

## 4.3 Comparison Methods

In our experiments, we included the following common transition matrix and baseline methods as comparison: (1) VolMinNet [17], (2) PeerLoss [21] (3) BLTM [37], (4) PartT [33], (5) MEIDTM [6], as well as state-of-the-art methods for learning with noisy labels: (6) Co-teaching [10], (7) ELR+ [18], (8) DivideMix [14], (9) SOP and SOP+ [19], (10) PGDF [5], (11) CC [40], (12) LRA [3] with SimCLR as encoder similarly.

Table 2: Test accuracy on CIFAR-10N and CIFAR-100N.

| | CIFAR-10N | | | | | CIFAR-100N |
| --- | --- | --- | --- | --- | --- | --- |
| | Aggregate | Random 1 | Random 2 | Random 3 | Worst | Noisy |
| Co-teaching | 91.20±0.13 | 90.33±0.13 | 90.30±0.17 | 90.15±0.18 | 83.83±0.13 | 60.37±0.27 |
| ELR+ | 94.83±0.10 | 94.43±0.41 | 94.20±0.24 | 94.34±0.22 | 91.09±1.60 | 66.72±0.07 |
| DivideMix | 95.01±0.71 | 95.16±0.19 | 94.89±0.23 | 95.03±0.20 | 92.56±0.42 | 71.13±0.48 |
| SOP+ | 95.61±0.13 | 95.28±0.13 | 95.31±0.10 | 95.39±0.11 | 93.24±0.21 | 67.81±0.23 |
| PGDF | 95.35±0.12 | 94.95±0.21 | 94.78±0.34 | 94.92±0.28 | 94.22±0.29 | 67.76±0.35 |
| CC | 95.63±0.21 | 95.11±0.31 | 94.93±0.37 | 95.09±0.21 | 94.24±0.40 | 71.21±0.22 |
| LRA | 94.57±0.23 | 94.19±0.17 | 94.38±0.42 | 94.02±0.32 | 93.20±0.59 | 70.96±0.53 |
| DTM | **96.13±0.17** | **95.98±0.22** | **96.01±0.28** | **95.78±0.34** | **94.93±0.21** | **72.51±0.30** |

## 4.4 Experimental Results on Synthetic Datasets

We primarily validated our proposed method DTM against previous instance-based transition matrix methods on synthetic CIFAR-10/100 noise datasets. These methods mainly focus on estimating the transition matrix and some methods applicable to instance-dependent label noise. We performed 5 independent runs for each experimental configuration, and the average values and standard deviations of each experiment are presented in Table 1.

The results demonstrate that our proposed DTR method outperforms other methods of the same category across various noise rates. It is evident that traditional transition matrix methods for class-dependent noise as VolMinNet exhibit subpar performance when handling instance-dependent noise. While even advanced transition matrix methods for instance-dependent label noise such as BLTM, ParT and MEIDTM, still show significant gaps compared to our method.

Furthermore, as the noise rates increase, the test accuracy of existing transition matrix methods significantly decline. This is particularly pronounced in the case of CIFAR-100 with 50% instance-dependent noise (IDN) data, where all transition matrix methods achieve test accuracy below 60%. In contrast, our proposed DTR method achieves a remarkable test accuracy of 74.85%, showcasing its exceptional performance. That demonstrates relatively robust performance of DTM with only a slight decrease as the noise rate increases.

This experiment clearly demonstrates that there is a significant performance gap between previous transition matrix methods and other advanced techniques, such as CC and LRA, when dealing with instance-dependent noise problems. However, the experimental results indicate that our proposed method DTM, which incorporates the diffusion model into the estimation of the transition matrix, outperforms these advanced techniques, except for the case of 40% noise in CIFAR-100, where our method slightly underperforms CC. It is evident that by leveraging the diffusion modeling to estimate the transition matrix, we effectively incorporate the image's feature information, leading to a substantial improvement in the effectiveness of the transition matrix.

## 4.5 Experimental Results on Real-World Datasets

In addition to synthetic datasets, we also applied our method to real-world datasets and compared it with other state-of-the-art techniques for handling label noise problems. The results are presented in Table 2 and Table 3.

Table 3: Test accuracy on Clothing1M, Webvision and ILSVRC12.

| | Clothing1M | Webvision | ILSVRC12 |
| --- | --- | --- | --- |
| Co-teaching | 69.2 | 63.6 | 61.5 |
| ELR+ | 74.81 | 77.78 | 70.29 |
| DivideMix | 74.76 | 77.32 | 75.20 |
| SOP+ | 74.98 | 77.60 | 75.29 |
| PGDF | 75.19 | 81.47 | 75.45 |
| CC | 75.40 | 79.36 | 76.08 |
| LRA | 75.32 | 80.05 | 76.64 |
| DTM | **75.57** | **81.95** | **77.55** |

The results demonstrate that regardless of the type of noise labels, whether it is aggregated, random, or the worst-case scenario in CIFAR-10N, as well as in CIFAR-100N with more label categories, our method consistently achieves the best results in handling real-world noise. When dealing with large datasets like Clothing1M and complex image datasets like Webvision, DTM also performs comparably to other state-of-the-art methods.

Through extensive experiments on five real-world datasets and the rusults on synthetic datasets above, our method outperforms the LRA method, which also utilizes the diffusion model for label noise problems. The LRA method models label diffusion with fewer dimensional information and lacks the rationale of our method, which considers noise generation from a transfer probability distribution perspective. The experiments demonstrate that our method achieves better learning performance by effectively integrating the transition matrix with the diffusion model.

## 4.6 Ablation Study

Besides the aforementioned experiments, we conducted ablation studies on proposed DTM method to assess the importance of each component. Table 4 presents the comparative results under 20% and 40% instance-dependent noise rates, where "w/o" denotes "without". We conducted ablation experiments on three components of our method, they are diffusion module, pre-trained encoder module, and clustering module respectively. "w/o diffusion" indicates directly using the features extracted by the pre-trained model for the classification task with the transition matrix. "w/o pre-train" means not extracting features through self-supervised learning and directly utilizing the classification network with the diffusion model. "w/o clustering" indicates that the initial transition matrix used for the diffusion model is the same for all samples.

Table 4: Ablation study of DTR. The data in the table represents the test accuracy.

|  | CIFAR-10 | | CIFAR-100 | |
| --- | --- | --- | --- | --- |
|  | IDN-0.2 | IDN-0.4 | IDN-0.2 | IDN-0.4 |
| w/o pre-train | 90.52 | 83.61 | 66.17 | 61.79 |
| w/o clustering | 92.25 | 88.35 | 71.93 | 66.47 |
| w/o diffusion | 93.74 | 91.66 | 79.82 | 73.51 |
| DTR | **95.90** | **94.82** | **82.04** | **78.56** |

From the results in Table 4, it can be observed that regardless of which component of diffusion module, pre-trained encoder module and clustering module is missing, the performance is consistently weaker compared to the original DTM. This indicates that each module plays a crucial role in our method. Our approach effectively combines the transition matrix, diffusion model, and pre-trained feature extraction, leading to significant improvements.

## 5 Conclusion

In this paper, we propose a method that models the transition matrix using diffusion models, incorporating the feature information extracted by a pre-trained encoder into the estimation of the transition matrix. This approach enables the model to handle instance-dependent label noise with a wider range of applicability. Experimental results on both synthetic and real-world noisy datasets demonstrate the effectiveness of our proposed method.

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
