# OpenReview forum: "Estimating Transition Matrix with Diffusion Models for Instance-Dependent Label Noise"
_NeurIPS.cc/2024/Conference — Submitted to NeurIPS 2024_

### Official Review · Reviewer_YAFq · 2024-07-12

**Soundness:** 1
**Presentation:** 2
**Contribution:** 2
**Rating:** 3
**Confidence:** 4

**Summary:**

This paper deals with the problem of supervised learning from noisy labels, where the label noise is modeled using instance-dependent label transition probability matrix. Mainly, this work attempts to leverage conditional diffusion model in order to obtain a generative model of transition matrix conditioned on the sample features. To that end, this work first generate pseudo paired samples $( x_i, T_i )_{i=1}^N$ using existing method (VolMinNet). Secondly, a conditional diffusion model is trained that generates $T_i$ given $x_i$. Finally, the classifier is trained taking into consideration the estimated transition matrix from the diffusion model.

**Strengths:**

1. The problem considered is of interest to the broad ML community
2. Adequate experimental settings, baselines, and ablations are provided for numerical validation.
3. The attempt to apply diffusion model is novel.

**Weaknesses:**

1. The technical soundness of the proposed method is questionable. Essentially, the proposed method trains a conditional diffusion model using paired samples $(x_i, T_i)$. If we consider the true transition matrix as $T(x)$ for a sample $x$, then the idea of the proposed method is to train a conditional generative model $p( T(x) | x )$. There are several issues with this attempt and the proposed implementation:
   (a) The authors use pseudo transition matrix $T_i$ generated from a sample-independent method (VolMinNet). $T_i$ only depends upon the cluster assignment of $x_i$. The diffusion model, at best, can approximate the conditional distribution $p( T_i | x_i )$. This has no clear relation to $p(T(x) | x)$. Therefore, in principle, the transition matrix generated by the trained diffusion model cannot be better than that returned by VolMinNet.
   (b) Second, the transition matrix is modeled as a deterministic function of sample, i.e., only one $T(x)$ exists for a given $x$. Therefore, it does not make sense to learn a generative model for $p(T(x) | x)$, since it is a degenerate distribution (probability of all other matrices should be zero except the true $T(x)$).

2. Another hint at why the proposed method should be limited by the pseudo paired sample distribution is that the diffusion model training part (which is ultimately used as transition matrix estimator) does not require available noisy labels. Hence, no extra information can be extracted about the true transition matrix $T(x)$ beyond the information captured by the pseudo paired samples $(x_i, T_i)$.

2. It is unclear where the performance gain in empirical results is coming from. The manuscript does not provide any intuitive or theoretical explanation to justify the quality of their estimator. Moreover, no rationale for the algorithm design is provided.

**Questions:**

Please provide a rebuttal for each of the points in weaknesses section.

**Limitations:**

Limitations are not adequately discussed.

---

### Official Review · Reviewer_87Qq · 2024-07-16

**Soundness:** 1
**Presentation:** 1
**Contribution:** 1
**Rating:** 2
**Confidence:** 4

**Summary:**

This paper focuses on the estimation of the transition matrix with instance-dependent label noise. They used a diffusion model for this estimation. By applying a diffusion process to the transition matrix, the diffusion model is trained to generate transition matrices from a prior distribution. The instance-wise generated transition matrix is then used to train the classifier with a forward cross-entropy loss. The improvement of the method is demonstrated by experiments on benchmark and real-world datasets.

**Strengths:**

The instance-dependent label noise scenario is a challenging task.

**Weaknesses:**

* The reason for generating the transition matrix using a diffusion model is unclear.
  * The instance-dependent transition matrix is the target to be estimated, but it is uncertain what role training a diffusion model to generate the transition matrix without a fixed target.
  * In addition, as mentioned by the authors, the transition matrix must be satisfied: the entries are greater than 0, the row sum is to be 1, and the diagonal entry is typically the largest. However, these considerations have not been taken into account in the construction of the diffusion process. Although a transformation method is proposed in Section 3.4, there is no discussion of how this affects the training of the diffusion model.

* Pre-trained features are fed into the diffusion network, but their impact on the diffusion process has not been analysed. This could be seen as providing additional conditional information during the diffusion process, implying that this diffusion model might be a conditional diffusion model. It would be better to discuss these consideration.

* In Algorithm 3, it appears that the diffusion model is trained in order to generate the initialized $T_i$. I wonder if the desired training is for the initialized $T_i$ to be generated perfectly as is. This could lead to a transition matrix that might not contain instance-dependent information, raising questions about the mechanism by which diffusion training introduces variance.

* The diffusion training seems to take a considerable amount of time, which needs to be analysed. If it takes a long time, the performance improvement may not be significant in comparison.

**Questions:**

Please see the Weaknesses part.

**Limitations:**

They mentioned the limitations only briefly in the experimental section. I have noted additional limitations that I perceive in the Weaknesses part.

---

### Official Review · Reviewer_s31f · 2024-07-17

**Soundness:** 1
**Presentation:** 2
**Contribution:** 1
**Rating:** 3
**Confidence:** 5

**Summary:**

In this work, the authors proposed an approach to estimate the instance-dependent transition matrix in order to reliably learn from noisy labels. The idea is to use a condition diffusion model to estimate the transition matrix by using the pretrained extracted image features as the conditions. Once the transition matrices are estimated, the classifier is learned through the corrected cross entropy loss. Experiments are presented to compare the performance of the approach with other baselines using both synthetic and real noisy datasets.

**Strengths:**

The paper is easy to read and notations are clearly stated

**Weaknesses:**

The main weakness is the lack of support and discussion in substantiating the idea. Experiments are insufficient to support the claims.

**Questions:**

Questions:

1.	A key concern is that estimation of the transition matrix is heavily dependent on the initializations given to the diffusion model learning. The diffusion model intuitively tries to approximate the distribution of its inputs through its forward and reverse process. In the traditional setting, the original image features is the input. But in your case, the initializations estimated through clustering and volmin optimization are the inputs. This part is quite unclear how does it help learn the true instance-dependent transition matrices.

2.	In the experiments, in Table 4, I do not see the ablation study with just using the initialized transition matrix and training the classifier, which is important to see the effect of the diffusion model-based learning for the TM. The ablation study corresponding to “w/o diffusion” says that it is using the pre-trained model.

3.	Experiment results all look good compared to the baselines, but I do not see any clear intuition/discussion to substantiate this idea of instance-dependent transition matrix estimation

**Limitations:**

No limitations are discussed

---

### Decision · Program_Chairs · 2024-09-25

**Decision:**

Reject

**Comment:**

This work proposes to tackle the noisy label learning problem by modeling instance-dependent label transition matrices using a diffusion model.

All reviewers expressed their dissatisfaction with the lack of motivation of using a diffusion model to approximate the label transition matrix. The reviewers are also not convinced by the soundness of the approach. The training method using pseudo transition matrices and the initialization methods also raised concerns. The authors did not respond in the rebuttal phase.